# The Use of Abdominal Ultrasound to Improve the Cryptorchidectomy of Pigs

**DOI:** 10.3390/ani12141763

**Published:** 2022-07-09

**Authors:** Alice Carbonari, Edoardo Lillo, Vincenzo Cicirelli, Raffaele Luigi Sciorsci, Annalisa Rizzo

**Affiliations:** Department of Veterinary Medicine, University of Bari ‘Aldo Moro’, 70121 Bari, Italy; alice.carbonari@uniba.it (A.C.); edoardo.lillo@uniba.it (E.L.); raffaeleluigi.sciorsci@uniba.it (R.L.S.); annalisa.rizzo@uniba.it (A.R.)

**Keywords:** swine, ultrasound examination, cryptorchidectomy, pain

## Abstract

**Simple Summary:**

Cryptorchidism is the failure of one or both the testes to descend in the scrotum. In pigs, the incidence of cryptorchidism varies from 4% to 12%, and the most frequent localisation of retained testes is the abdominal region. In other species, transabdominal ultrasound is the most reliable diagnostic tool to localize the retained testis. This study aimed to evaluate the value of ultrasound in correctly identifying the location of retained testes, to improve the cryptorchidectomy in pigs. The ultrasound examination reduced the duration of anaesthesia, surgery, and postoperative pain. Indeed, the technique used is safe, effective, and rapid. To locate the testis, it should recognise the echo structure of the testicle and the mediastinum is well always recognisable for its hyperechogenicity. The routine use of the ultrasound examination is considered a useful alternative before the cryptorchidectomy in the pig because it allows the exact location of the retained testicle, ensuring reduced manipulation of the viscera.

**Abstract:**

This study aimed to describe a technique to locate retained testes in pigs by means of ultrasound examination and thereafter proceed with cryptorchidectomy. Fifty-two monolateral cryptorchid pigs were enrolled. After anaesthesia, 42 pigs (Group U) underwent ultrasound examination before cryptorchidectomy, and 10 pigs (Group C) were used as control group without ultrasonography. The total duration of anaesthesia, ultrasound examination, and surgery was evaluated. In 10 pigs of Group U and 10 pigs of Group C, the heart rate (HR), respiratory rate (RR), and body temperature (T) were monitored to assess intraoperative pain response. An operator used the Piglet Grimace Scale (PGS) to assess postoperative pain. In Group U, the total time required for anaesthesia and surgery was less than Group C. No intra- or postoperative complications were reported in both groups. For HR, RR, and T, no statistically significant differences were observed. During the postoperative pain assessment, the highest pain scores were recorded at T6 (6 h after surgery). Ultrasound examination was found to be a valid technique for locating the retained testis in the abdomen in cryptorchid pigs and to reduce the duration of the surgery.

## 1. Introduction

Cryptorchidism is the failure of one or both the testes to descend in the scrotum. In pigs, the migration of the testis is physiologically complete at the time of birth [1,2]. Numerous causes such as oestrogenic environmental agents, epigenetic alterations, and gene mutations have been implicated in this pathology [3,4,5,6,7,8]. Regardless of the cause, the alteration in the migration of the testis can induce a block of the testis in the abdominal cavity, inguinal canal, or subcutaneous (external to the abdominal wall) location [1]. In pigs, the incidence of cryptorchidism varies from 4% to 12% and the most frequent localisation of retained testes is the abdominal region [9]. Several problems are correlated with cryptorchidism. Clinical problems such as post-puberal pathologies correlated with cryptorchidism are altered spermatogenesis in scrotal testes, atypical concentrations of reproductive hormones, and testis tumours [1]. In addition, cryptorchidism has a negative economic impact because the presence of testes results in boar taint, which is a bad smell or taste of meat of these animals [10]. Regulation 625/2017/EC states that if meat has organoleptic anomalies, in particular an intense smell, because of androstenone and skatole, it is unfit for human consumption. Androstenone is produced in the testis and has a urine-like smell. In addition, it has been found to affect skatole metabolism in the liver. Skatole, formed from tryptophan, is produced in the colon and has a faecal-like odour [11,12]. For these reasons, undescended testis should be removed. Surgery can be performed using a paramedian approach, parainguinal laparotomy, or incision at the cranial commissure of the external inguinal ring (non-invasive inguinal approach) [9,13,14]. In these studies, the exploration of the abdomen to locate the testes was performed with a gloved finger. In other species, such as equine, canine, feline, and human, transabdominal ultrasound is the most reliable diagnostic tool to localize the retained testis [15,16,17,18,19,20,21,22]. In particular, in dogs, Felumlee et al. (2012) found that ultrasound has a 100% positive predictive value for all retained testes and the sensitivity of ultrasound was 96.6% for abdominal and 100% for inguinal testes. In horses, the sensitivity of transabdominal ultrasonography was 97.6% and its specificity was 100% [15]. In humans, the sensitivity and specificity of ultrasonography was 85% and 25%, respectively [20]. This study aimed to evaluate the value of ultrasound in correctly identifying the location of retained testes in pigs to perform cryptorchidectomy in a simple and quick manner without risks. To assess the efficacy of this method, the duration of all performed procedures (anaesthesia, ultrasound, and cryptorchidectomy), intra- and postoperative complications, were evaluated. Moreover, pain evaluation was performed before and after surgery using the Piglet Grimace Scale (PGS) [23]; the heart rate (HR), respiratory rate (RR), and body temperature (T) were monitored and potential variations in these parameters were analysed to assess pain. Our hypothesis is that, as in other species, the use of ultrasonography is useful for identification and location of the abdominal testis, precluding complications associated with abdominal exploration and reducing pain.

## 2. Materials and Methods

### 2.1. Study Design

This was a randomized clinical research study.

### 2.2. Animals

This study enrolled 52 commercial hybrid pigs. All pigs were reared in farms in South Italy, were 3–6 months old, weighed 30–80 kg, and were referred by the owner or farm veterinary for suspected cryptorchidism. In all pigs, the testis in the scrotum was removed, by the breeder or other trained personnel, by seventh day of life, during routine castration at the farm. Each animal was clinically and physically examined. Only pigs with complete abdominal retention of one testis (monolateral cryptorchidism) and without other pathologies were enrolled in this study. The genital apparatus was examined. Inguinal regions were palpated to exclude the *retentio inguinalis* and the scrotal area was inspected to identify the previous castration scar of the removed testicle. This helped the operator to determine the side of the undescended testicle. All pigs were fasted 12 h prior to surgery, and the access to water was removed 2 h prior to cryptorchidectomy. The animals were premedicated with 4 mg/kg azaperone (Stresnil, Elanco Italia S.p.A.; 40 mg/mL) by deep intramuscular (IM) injection. After 15 min, anaesthesia was induced by IM 10 mg/kg ketamine (Anaestamine, PH Farmaceutici; 100 mg/mL) and 0.08 mg/kg detomidine hydrochloride (Domosedan, Vètoquinol Italia Srl; 10 mg/mL). This protocol was in accordance with that described by Heinonen et al. (2009) [24]. After approximately 5 min, the auricular vein was catheterised under aseptic conditions, using an 18 G venous catheter (Deltaven), for possible subsequent anaesthetic boluses performed with detomidine hydrochloride (0.02 mg/kg) and ketamine (2.5 mg/kg). Duration of anaesthesia was recorded, from the time to unconsciousness, evidenced by lateral recumbency, head down, and lack of reaction when manipulating or moving their body, and absence of the palpebral reflex, to the time to first spontaneous movement, standing position, and the number of attempts to stand. After anaesthesia, 42 pigs (Group U) underwent ultrasound examination before cryptorchidectomy, and 10 pigs (Group C) were used as control group, without ultrasonography. Operative time (ultrasound examination and surgery time) and intraoperative and postoperative complications were recorded. Among the 52 pigs, 20 (10 belonging Group U and 10 of Group C) were from the same farm and were considered for pre- and postoperative pain evaluation. The free software, Java applets for power and sample size, was used to determine the minimum sample size [25].

### 2.3. Ultrasound Examination 

Ultrasound examination was always performed by the same operator. After inducing anaesthesia, the pigs were placed in the dorsal decubitus position. The abdominal and inguinal regions were trichotomized and alcohol was used to clean cutaneous sebum from the skin. Thereafter, an ultrasound examination (SonoSite MicroMaxx, Bothell WA, USA) was performed with a 7.5 MHz linear probe to confirm cryptorchidism and to locate the testis. The ultrasound gel was applied to the probe to increase the conductivity of ultrasound waves through the dermis and underlying tissues and remove the resistance opposed by air to their propagation. The ultrasound probe was placed on the abdominal wall at the level of the internal inguinal ring and moved cranially to locate the retained testicle. The undescended testicle was located based on the peripheral hyperechogenicity of the albuginea, homogeneous parenchymatous echo-texture typical of the testis, and hyperechogenicity of the mediastinum (Figure 1).

The time required for the ultrasound examination was calculated as the time from the placement of the probe on the pig to recognition of the testis. 

### 2.4. Surgical Approach

In Group U, the point at which the testis was ultrasonographically located was marked with a Sharpie^®^ W10 black permanent marker. In Group C, the surgical area was identified blind at the caudal part of the abdomen, cranial to the scrotum, 3 to 5 cm lateral to the midline. Thereafter, the surgical area was scrubbed with three alternating passages of alcohol and 10% iodine povidone. The same team of surgeon performed all the procedures. Various layers of the abdominal region, skin, subcutaneous tissue, muscular planes, and peritoneum were incised with tissue scissors and a scalpel blade. In Group U, the testicle was immediately found, while in Group C the exploration of the abdomen was performed with a gloved finger directed first toward the inguinal ring and then toward the caudal pole of the ipsilateral kidney. In both groups, once the testis was detected, it was externalised and the spermatic cord was clamped with Klemmer forceps. At the upstream of the forceps, a transfixing ligature and a simple ligature were performed. The second forceps was placed perpendicular to the first and the testicle was removed by twisting. After testicle removal, the surgeon ensured no bleeding and performed a continuous everting entangled suture with single reinforcement stitches at the muscle plane and sutured the skin and subcutis with horizontal U-stitches. The suture thread used was USP 3-4 absorbable (Surgicryl^®^-Steinerberg–Belgium). The duration of surgery, from skin incision to the last stitch on the skin, was calculated. Retained testes were not histologically examined. Intramuscular injection of 40.000 UI/kg benzilpenicilline and 50 mg/kg dihydrostreptomicine (Repen, Fatro S.p.A., Bologna, Italy; 200.000 UI + 250 mg/mL) were administered 2 h before and at the end of surgery to provide antibiotic coverage. Intramuscular injection of 0.2 mg/kg dexamethasone (Fatrocortin, Fatro S.p.A.; 1 mg/mL) was administered at the end of surgery as anti-inflammatory coverage and for gluconeogenesis stimulation.

### 2.5. Pain Assessment

In 10 pigs of Group U and 10 pigs of Group C, pain was assessed during pre-, intra-, and postoperative procedures by the same operator.

***Pre-operative evaluation.*** Preoperative pain assessment was performed at the farm (T0). The operator assigned a pain score based on the PGS. The PGS comprised three facial action units (FAUs), each of which was scored. The position of ears (0 = keep forward; 1 = moved backward; 2 = drawn back from forward position) and the position of the grump (0 = lack of snout bulge and cheek tension; 1 = evidence of skin bulge on the snout; 2 = visible wrinkles caudal to the snout) were evaluated separately based on a score ranging from 0 to 2, and the closing of the eyes was evaluated based on a score ranging from 0 to 1 (0 = not present and 1 = present) [23].

***Intra-operative evaluation*** Vital parameters such as HR, RR, and T were measured at the following five different timepoints to assess intraoperative pain:-Surgical preparation (after placing the animal on the operating table and inducing anaesthesia) (T1);-Skin incision (T2);-Manipulation of the testicle (T3)-Resection of the testis and release of residual stump of the spermatic cord (T4);-End of surgery (after applying the last simple stitch on the skin) (T5).

Non-invasive monitoring instrumentation (M9000 Vet–Foschi srl) was attached to the pig’s chest to measure heart and respiratory rate. The rectal temperature was measured with a digital thermometer.

***Postoperative evaluation.*** Postoperative pain assessment was performed at 6 h (T6), in accordance with Vullo et al. (2020) [23], and 24 h (T7) after the end of the surgery using the PGS. In case of a PGS score of ≥2/5, rescue analgesia was performed by administering 300 mg/100 kg IM ketoprofen (Zooketo, Bayer S.p.A.; 100 mg/mL).

### 2.6. Statistical Analysis

Compiled forms were entered into a database created with an Excel spreadsheet and data analysis was performed using R Studio software (version 2022.02.0). Continuous variables were described as media (standard deviation [SD]). The skewness and kurtosis test were used to evaluate the normality of continuous variables; a normalization model was set up to normalize those not normally distributed. The T Student test was performed to compare the duration of anaesthesia and surgery between two groups. The ANOVA for repeated measures test was used to compare continuous variables between groups and detection time. For all tests, a 2-sided *p*-value < 0.05 was considered statistically significant.

**Figure 1 animals-12-01763-f001:**
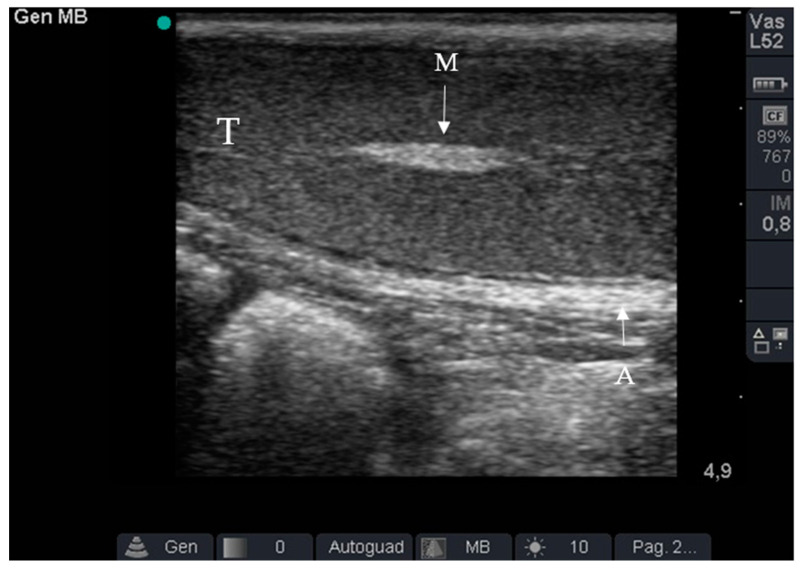
Ultrasound of the undescended testicle. Note peripheral hyperechogenicity of the tunica albuginea (A), the homogeneous parenchymatous echo-texture typical of the testis (T), and the hyperechogenicity of mediastinum (M).

## 3. Results

Among the 52 unilaterally cryptorchid pigs enrolled in the study, 49 (94%) pigs had the undescended testicle in the abdomen on the left side. The ultrasound examination was easy to perform. In 36 pigs, the undescended testicle was identified based on the hyperechogenicity of the mediastinum and the echo-texture typical of the testis. However, ultrasound examination was more complicated in 6 pigs because the echotexture of the testis was not easily recognisable, although the hyperechogenicity of the mediastinum was always noticeable (Figure 2).

The time for ultrasound examination is shown in Table 1. In all pigs of Group U, the undescended testis was surgically found to be present at an ultrasonographically located position. Hence, no manipulation of the intestine was required, and testis externalisation was easy. In both groups, the duration of anaesthesia and surgery is shown in Table 1. In all pigs of Group C, a second bolus of anaesthesia was necessary.

In both groups, no intra- and postoperative complications were reported. The results of HR, RR, and T evaluation performed intraoperatively were presented in Figure 3, Figure 4 and Figure 5. The HR, RR, and T values are within the range of species [26,27]. Regarding the trend of HR, in Group C, it was possible to observe a peak at T4, followed by a reduction at T5, while in Group U, the trend was uniform during the inspected times. From the analysis of ANOVA for repeated measures, no statistically significant differences were observed in the comparison of HR between groups (*p* = 0.520), interaction between time and group (*p* = 0.540), and between times (*p* = 0.737). As RR, in both groups, the values increased at T4 and T5. From the analysis of ANOVA for repeated measures, no statistically significant differences were observed in the comparison of RR between groups (*p* = 0.485), interaction between time and group (*p* = 0.971), and between times (*p* = 0.427). The T values decreased from T1 to T3, followed by an increase until at T5. From the analysis of ANOVA for repeated measures, no statistically significant differences were observed in the comparison of T between groups (*p* = 0.114) and interaction between time and group (*p* = 0.937), while it is observed in the comparison between times (*p* = 0.002).

The PGS values varied from T0 to T7 (Figure 6). The maximum PGS value was observed at T6, and after then the score decreased until T7. From the analysis of ANOVA for repeated measures, statistically significant differences were observed in the comparison of the Piglet Grimace Scale between groups (*p* = 0.0002), times (*p* < 0.0001), and interaction between time and group (*p* = 0.035). At T6, all pigs who underwent pain evaluation had a PGS score of >2/5 and were administered an IM injection of ketoprofen. Given the uniformity of PGS values at T6 (>2/5) in the 10 pigs evaluated, analgesia by administration of ketoprofen IM was applied to all animals enrolled for the study 6 h after the end of surgery.

## 4. Discussion

In this study, testis retention was observed to be mainly on the left side in the abdomen, which is consistent with that reported by Skelton et al. (2021) [9]. Anatomically, in porcine species, the left kidney is more cranial than the right kidney [28], and therefore the pathway during the testicular descent phase is greater. This could explain the higher prevalence of retained testes on the left side in the abdomen in swine species. The use of ultrasound examination before surgery yielded excellent results. This diagnostic examination is used frequently in different species (human, equine, canine, feline) but in pigs there are no data on this method. Ultrasound is a sensitive and specific technique to locate retained testes and it has value because it is non-invasive and facilitates planning of the correct surgical procedure [17]. The ultrasonography use reduced the manipulation of abdominal organs (mainly the intestine) and eventually postoperative complications. It is known that the manipulation of the intestine during open surgery results in oxidative stress in the intestinal mucosa, leading to permeability alterations with bacterial and endotoxin translocation. Surgical manipulation also induces a significant expression of the proinflammatory cytokines, with consequence ascites and peritoneal adhesion [29,30]. Moreover, the duration of the anaesthesia and surgery, and therefore the exposure time of the abdomen to the external environment, was considerably reduced in Group U compared with that of Group C. The mean ± SD surgery time was 15.39 ± 2.53 min; thus, the mean (± SD) surgery time plus the mean (±SD) time required for ultrasonography (6.72 ± 2.37 min) in this study was considerably lower than the mean time required for surgery alone. The duration of surgery in Group C was similar (32 ± 10.2 min) to a previous study by Skelton et al. [9]. This makes it possible to reduce the duration of anaesthesia, and therefore the associated risks. The technique used in this study is safe, effective, and rapid, although extremely operator dependent. The veterinarian who performs the ultrasound examination should know the echo structure of the testicle very well to be able to quickly recognise and locate it. This condition proves to be of fundamental importance, especially when the undescended testicle is degenerated. In this study, in six pigs, the time required for ultrasound examination was longer because of the testes degeneration. However, the mediastinum was well always recognisable for its hyperechogenicity, and this represents an important parameter for identifying the testicle. During the surgery, HR, RR, and T were monitored to evaluate anaesthesia and pain. The variations in the tested parameters were within the range for the species [26,27]. As HR, a peak is evident at T4 in Group C. This increase, although not statistically significant, is due to a superficialization of anaesthesia, and for this, all pigs underwent another bolus of anaesthesia. The superficialization of anaesthesia was also evident by higher RR values of Group C than Group U. The T values showed a reduction from T1 to T3, which may be related to the hypothermic effect of anaesthesia, as after this there was an increase of T. However, the recorded T values were physiological for swine. The pre- and postoperative pain assessment using the PGS highlighted that all pigs experienced extreme pain only at T6 (PGS: > 2/5), requiring IM ketoprofen injection. At 24 h after surgery (T7), the pain scores decreased considerably in all pigs. The statistical comparison showed a statistically significant difference between groups, and fast recovery in Group U may be because less manipulation of abdominal organs was performed to locate the undescended testis and the short duration of the entire surgery. The ultrasound examination was found to be an extremely safe and fast technique and a key feature of field surgeries in pigs. The reduced duration of surgery led to minor pain in postoperative assessment. Therefore, performing ultrasound examination before the surgery allows the incision in the abdominal wall at the exact location of the retained testicle, ensuring reduced manipulation of the viscera.

## 5. Conclusions

Routine use of the abdominal ultrasound is considered desirable for cryptorchidectomy of pigs, which can be performed by reducing the intraoperatory manipulation. The ultrasound decreases the procedure period, improving the postoperative recovery. These findings may provide the foundations for further investigations about the ultrasound as collateral examination in porcine medicine.

## Figures and Tables

**Figure 2 animals-12-01763-f002:**
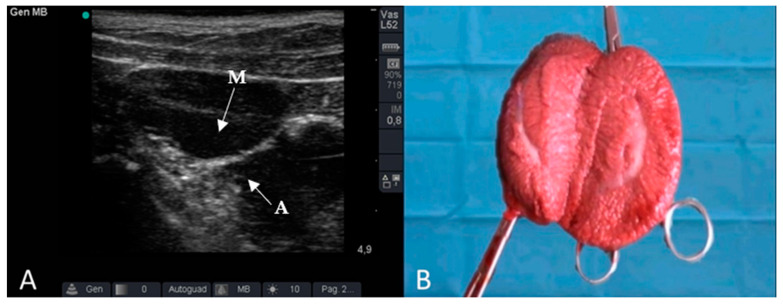
Ultrasound (**A**) and macroscopic features (**B**) of the undescended testicle. (A) tunica albuginea, (M): mediastinum.

**Figure 3 animals-12-01763-f003:**
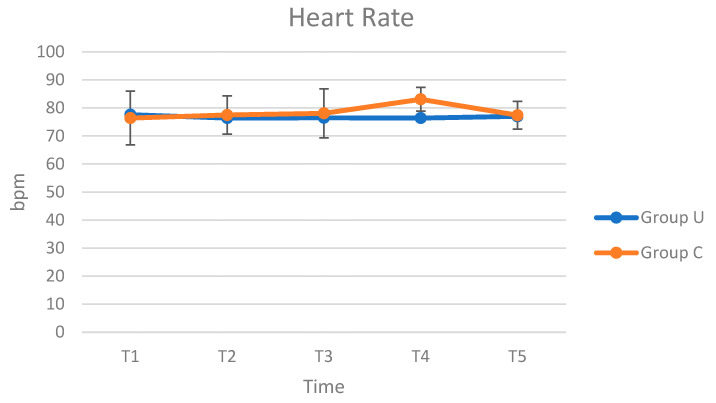
Average HR (bpm) values in Group U (pigs underwent ultrasound examination before cryptorchidectomy) and Group C (control group, without ultrasonography), at times T1 (surgical preparation, after placing the animal on the operating table and inducing anaesthesia), T2 (skin incision), T3 (manipulation of the testicle), T4 (resection of the testis and release of residual stump of the spermatic cord), and T5 (end of surgery, after applying the last simple stitch on the skin).

**Figure 4 animals-12-01763-f004:**
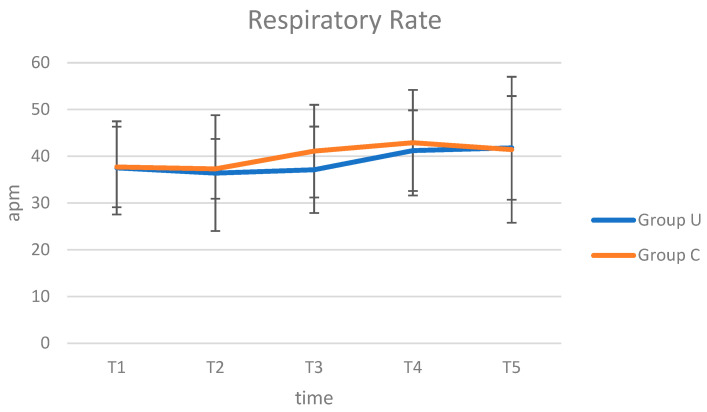
Average values of RR (apm), values in Group U (pigs underwent ultrasound examination before cryptorchidectomy) and Group C (control group, without ultrasonography), at times T1 (surgical preparation, after placing the animal on the operating table and inducing anaesthesia), T2 (skin incision), T3 (manipulation of the testicle), T4 (resection of the testis and release of residual stump of the spermatic cord), and T5 (end of surgery, after applying the last simple stitch on the skin).

**Figure 5 animals-12-01763-f005:**
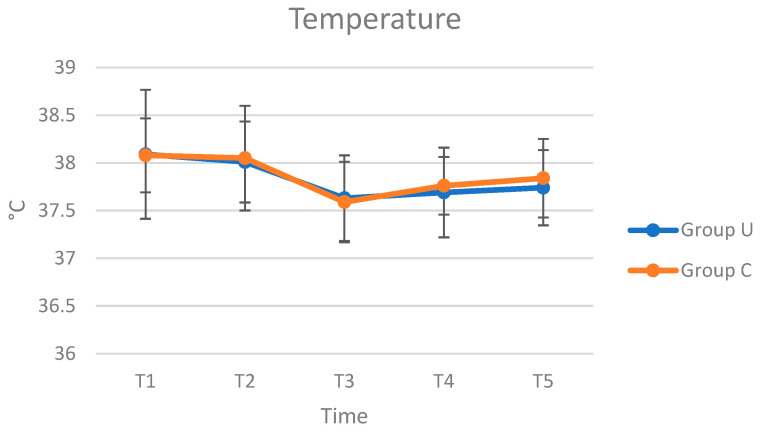
Average values of T (°C), in Group U (pigs underwent ultrasound examination before cryptorchidectomy) and Group C (control group, without ultrasonography), at times T1 (surgical preparation, after placing the animal on the operating table and inducing anaesthesia), T2 (skin incision), T3 (manipulation of the testicle), T4 (resection of the testis and release of residual stump of the spermatic cord), and T5 (end of surgery, after applying the last simple stitch on the skin).

**Figure 6 animals-12-01763-f006:**
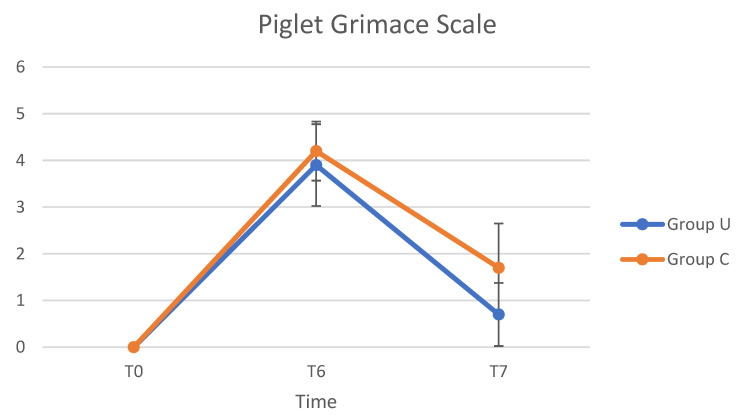
Average values of the Piglet Grimace Scale, in Group U (pigs underwent ultrasound examination before cryptorchidectomy) and Group C (control group, without ultrasonography), at times T0 (preoperative pain), T6 (6 h after the end of the surgery), and T7 (24 h after the end of the surgery).

**Table 1 animals-12-01763-t001:** Duration of anaesthesia, ultrasonography, and surgery in Group U (pigs underwent ultrasound examination before cryptorchidectomy) and Group C (control group, without ultrasonography). In column: a,b: *p* < 0.05; c,d: *p* < 0.001.

Groups	Duration of Anesthesia (minutes)	Duration of Ultrasonography (minutes)	Duration of Surgery (minutes)
Group U	35.02 ± 6.35 a	6.52 ± 2.37	15.39 ± 2.53 c
Group C	45.35 ± 10.36 b	/	35.00 ± 12.5 d

## Data Availability

The data presented in this study are available on request from the corresponding author.

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
