# Peer review of "The Use of Abdominal Ultrasound to Improve the Cryptorchidectomy of Pigs"

_animals, 2022, doi:10.3390/ani12141763_

Round 1

Reviewer 1 Report

The dangers of cryptorchid piglets are many. If breeding boars, it cannot be used for breeding. If it is a commercial pork pig, it is directly fattened without special surgical castration, which not only grows slowly, consumes a lot of material, and has poor meat quality, but also is inconvenient for mixed breeding in feeding management, which is not conducive to management.

The growth rate is fast, the fat is easy to gain, the feed reward is high, and the meat is delicious. This article provides a method for cryptorchidism surgery under ultrasound. Major revisions:

1. All data are not subject to data statistics and analysis.

2. There is no significant difference analysis in Figure 3-6 and bar

3. Figure 3-6 remove the border of the figure 4. Table 1 becomes a three-line table

Author Response

Reviewer 1

The dangers of cryptorchid piglets are many. If breeding boars, it cannot be used for breeding. If it is a commercial pork pig, it is directly fattened without special surgical castration, which not only grows slowly, consumes a lot of material, and has poor meat quality, but also is inconvenient for mixed breeding in feeding management, which is not conducive to management.

The growth rate is fast, the fat is easy to gain, the feed reward is high, and the meat is delicious. This article provides a method for cryptorchidism surgery under ultrasound.

Thank You for your comments and suggestions.

Major revisions:

  1. All data are not subject to data statistics and analysis.

We added the statistical analysis of data inserted in table 1.

  1. There is no significant difference analysis in Figure 3-6 and bar

We added the bars of standard deviation in all figures.

  1. Figure 3-6 remove the border of the figure 4. Table 1 becomes a three-line table

We modified the figures and table.  

Reviewer 2 Report

Dear authors,
The manuscript “The use of abdominal ultrasound to improve the cryptorchidectomy of pigs” aimed to
evaluate the value of ultrasound in correctly identifying the location of retained testes in pigs, to perform cryptorchidectomy in a simple and quick manner without risks. It is a very interesting study, as the importance of ultrasound is highlighted under field condition, before the cryptorchidectomy in the pig, because allows the exact location of the retained testicle, ensuring reduced manipulation of the viscera.
The technique is usually used in human and in horses, dogs and cats, but in pigs is not routinely used. So in this study, it is highlighted the importance of ultrasound also in this species. The manuscript is well explained, with sections presented in a balanced and coherent way. Introduction is properly documented, and the data are properly discussed. The experimental study is well set up.

The main objective of the manuscript “The use of abdominal ultrasound to improve the cryptorchidectomy of pigs” was to evaluate of the use of ultrasound examination for correctly identify the location of retained testicle in pigs, to perform cryptorchidectomy easily and quickly,
without risks. This is a very interesting, original, and relevant study in swine’s medicine, as it highlighted the importance of ultrasonography under field conditions, before cryptorchidectomy. The technique is usually used in human and in horses, dogs, and cats, but in pigs is not routinely used. In this study, it is highlighted the importance of ultrasound also in this species. Ultrasonography allows the exact location of the retained testicle, ensuring reduced manipulation of the viscera and reducing the time of surgery. Due to these characteristics, there is an improvement of post-operative recovery without complications. In the literature, cryptorchidectomy surgery is
described in several scientific articles, but in none of them the ultrasound is used as a technique to detect the retained testicle: in fact, the surgery times turn out to be longer than those described in this manuscript regarding the group of animals undergoing ultrasound examination. The manuscript
is well explained, with sections presented in a balanced and coherent way. Introduction is properly documented, and the data are properly discussed. The experimental study is well set up. Conclusions from the experimental study results meet the objectives initially set, they are well explained, and are consistent with the results obtained. The references are well reported, meeting
the journal guidelines for authors, and are relevant to what is written in the manuscript. For these reasons, the manuscript can be accepted, after minor revision.

ï‚· Why pain scores were assessed for the 6 hours?
ï‚· What is the grump position? Elaborate.
ï‚· In Figure 2 (A and B), add the reference letters for the various structures of the testis.
ï‚· In Figures 3, 4, 5, 6 add the values of the Standard Deviation.
ï‚· The bibliography should be expanded.

ï‚· Why pain scores were assessed for the 6 hours?
ï‚· The bibliography should be expanded.

Author Response

Reviewer 2

Dear authors,
The manuscript “The use of abdominal ultrasound to improve the cryptorchidectomy of pigs” aimed to
evaluate the value of ultrasound in correctly identifying the location of retained testes in pigs, to perform cryptorchidectomy in a simple and quick manner without risks. It is a very interesting study, as the importance of ultrasound is highlighted under field condition, before the cryptorchidectomy in the pig, because allows the exact location of the retained testicle, ensuring reduced manipulation of the viscera.
The technique is usually used in human and in horses, dogs and cats, but in pigs is not routinely used. So in this study, it is highlighted the importance of ultrasound also in this species. The manuscript is well explained, with sections presented in a balanced and coherent way. Introduction is properly documented, and the data are properly discussed. The experimental study is well set up.

The main objective of the manuscript “The use of abdominal ultrasound to improve the cryptorchidectomy of pigs” was to evaluate of the use of ultrasound examination for correctly identify the location of retained testicle in pigs, to perform cryptorchidectomy easily and quickly,
without risks. This is a very interesting, original, and relevant study in swine’s medicine, as it highlighted the importance of ultrasonography under field conditions, before cryptorchidectomy. The technique is usually used in human and in horses, dogs, and cats, but in pigs is not routinely used. In this study, it is highlighted the importance of ultrasound also in this species. Ultrasonography allows the exact location of the retained testicle, ensuring reduced manipulation of the viscera and reducing the time of surgery. Due to these characteristics, there is an improvement of post-operative recovery without complications. In the literature, cryptorchidectomy surgery is described in several scientific articles, but in none of them the ultrasound is used as a technique to detect the retained testicle: in fact, the surgery times turn out to be longer than those described in this manuscript regarding the group of animals undergoing ultrasound examination. The manuscript is well explained, with sections presented in a balanced and coherent way. Introduction is properly documented, and the data are properly discussed. The experimental study is well set up. Conclusions from the experimental study results meet the objectives initially set, they are well explained, and are consistent with the results obtained. The references are well reported, meeting
the journal guidelines for authors, and are relevant to what is written in the manuscript. For these reasons, the manuscript can be accepted, after minor revision.

Dear Rewier, thank you for your revision.

Below are the responses to your comments.

  • Why pain scores were assessed for the 6 hours?

Regarding the postoperative pain assessment, it was performed based on the study conducted by Vullo et al 2020, as cited in the manuscript. In their work, the assessment was performed at 6 hours after the end of surgery.

  • What is the grump position? Elaborate.

Score 0 is conferred by referring to the lack of snout bulge and cheek tension.

Score 1 is assigned when there is evidence of skin bulge on the snout in response to increased cheek tension, while wrinkles caudal to the snout are not visible.

Score 2 is awarded when there is bulging of the skin on the snout in response to cheek tension and wrinkles caudal to the snout are clearly visible.

These parameters were used based on the study conducted by Vullo et al, 2020 regarding the Piglet Grimace Scale (PGS).

We added this information in manuscript.

  • In Figure 2 (A and B), add the reference letters for the various structures of the testis.

We added the reference letters in Figure 2.

  • In Figures 3, 4, 5, 6 add the values of the Standard Deviation.

We added the values of Standard Deviation in Figures 3, 4, 5 and 6.

  • The bibliography should be expanded.

We have provided for the expansion of the bibliography.

Round 2

Reviewer 1 Report

Accepted it.